# TABULAR DATA TO IMAGE GENERATION: BENCHMARK DATA, APPROACHES, AND EVALUATION

## ABSTRACT

In this work, we study the problem of generating a set of images from an arbitrary tabular dataset. The set of generated images provides an intuitive visual summary of the tabular data that can be quickly and easily communicated and understood by the user. More specifically, we formally introduce this new dataset to image generation task and discuss a few motivating applications including exploratory data analysis and understanding customer segments for creating better marketing campaigns. We then curate a benchmark dataset for training such models, which we release publicly for others to use and develop new models for other important applications of interest. Further, we describe a general and flexible framework that serves as a fundamental basis for studying and developing models for this new task of generating images from tabular data. From the framework, we propose a few different approaches with varying levels of complexity and tradeoffs. One such approach leverages both numerical and textual data as the input to our image generation pipeline. The pipeline consists of an image decoder and a conditional auto-regressive sequence generation model which also includes a pre-trained tabular representation in the input layer. We evaluate the performance of these approaches through several quantitative metrics (FID for image quality and LPIPS scores for image diversity).

## 1 INTRODUCTION

In recent years, conditional image generation has been one of the most important directions in the line of research for generative models, due to both its technical challenges and numerous potential applications of such technology. However, most of the works that investigates conditional generation of images consider images (Gatys et al., 2016) or text (Ramesh et al., 2021; 2022) as input. Based on this fact, it is natural to ask whether we can extend the input to other data types to discover the potential benefit of image generation models to a larger variety of domains. Hence, in this work, we study the possibility of generating images from a given tabular data, motivated by its promising capability to be applied in customer segmentation analysis for marketers and exploratory data analysis.

More specifically, we consider the following problem. Given a (tabular) dataset, or more generally a subset of rows and columns of the dataset[1], how can we automatically generate a set of high quality images that describe it? Additionally, the set of images generated from the dataset should characterize the key trends, patterns, and segments (clusters) in the data. Such image generation model would yield interesting possibilities. For instance, suppose we have a dataset of customers and the items they purchased, then instead of performing a thorough data mining that requires intensive technical expertise, an image that illustrates a specific segment of customers purchasing specific items already reveals valuable information about the underlying traits of consumer behaviors, which can be easily used in future targeted marketing campaigns. Hence, tabular-data-to-image generation can likely bring up interesting usage.

Tabular data to image generation has many important and practical applications. One important application is as a fundamental tool for exploratory data analysis. Consider a user that is interactively

---

[1]For convenience, the term dataset is used to refer to a subset of rows and columns from a dataset as well as the full dataset.

exploring a large dataset of interest using a standard visual analytics platform such as Tableau. In Figure 1, we provide an intuitive example demonstrating the utility of this problem for real-time interactive data exploratory and understanding. In this example, the user is exploring a large dataset using an interactive scatter plot matrix where they can select the data points of interest to explore by simply brushing over them. The user draws a rectangle around the data points (rows) and attributes (columns) of interest, which intuitively selects all such rows that satisfy the user-defined constraint. This selection then triggers a query to our data table to image generation model, resulting in the top-3 images being generated that are shown on the right in Figure 1. Most importantly, the user can immediately understand what the data points they selected actually represent by simply examining the intuitive set of images generated from the tabular data selected. They can then use the interface to select other groups of points representing a subset of rows in the tabular data, and immediately understand the essence of what they represent.

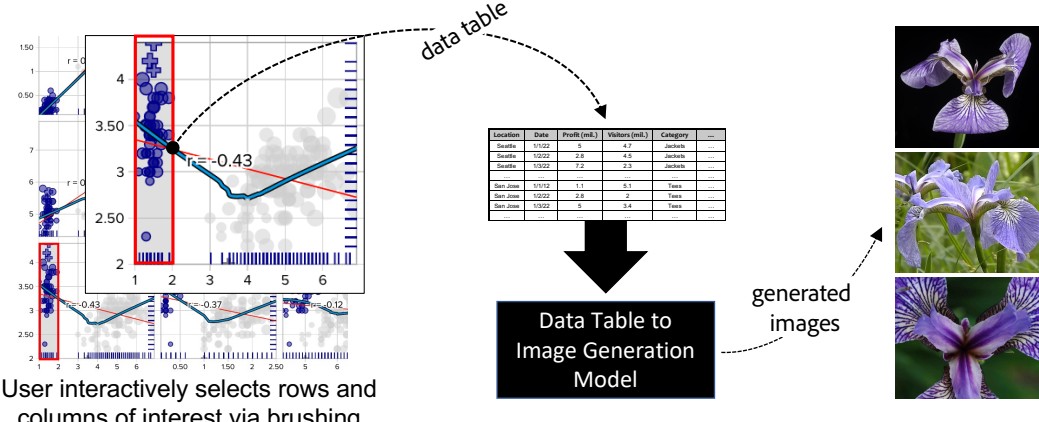

Figure 1: Tabular Dataset to Image Generation for Interactive Exploratory Data Analysis. See text for detailed discussion.

In this work, we address the problem of tabular-data-to-image generation by analyzing multiple possible approaches. To evaluate the performance of each of these approaches, we construct a benchmark dataset consisting of 300 tabular-data and images that are paired up based on a simple mapping rule. The possible approaches include directly combining existing implementations of DALL-E and GPT-3, and an end-to-end approach, in which we train a model similar to the architecture of DALL-E. We measure the performance of these approaches by computing the FID and LPIPS of the generated images with respect to the benchmark dataset, which captures the generated images' quality and diversity respectively.

The experiment results show that the approaches combining pre-existing models tend to generate images with better quality; however in terms of diversity, the end-to-end model we trained yields a better performance. Nevertheless, the qualitative results indicate that all proposed approaches still have significant room for improvement, and it is worth investigating if a better performance metric should be employed.

This work makes the following key contributions. First, we formally introduce the problem of image generation conditioned on tabular data input, which to the best of our knowledge has not been considered by any related works before. Second, we curate a benchmark dataset for training models specifically for the new problem studied in this work. Third, we propose and develop a few different approaches for solving it, and extensively evaluate them the quality and diversity of the images generated from such models. Furthermore, the framework proposed in this work is highly flexible, giving rise to a variety of other models that can be investigated in the future.

**Summary of main contributions**      The key contributions of this work are as follows:

- **Tabular Dataset to Image Generation Problem:** To the best of our knowledge, this work is the first to study the problem of generating images from a tabular dataset. Furthermore,

we also discuss a number of important applications for interactive exploratory data analysis and to better understand segments in an online marketing application.

- **Benchmark dataset:** To study this new and important task, we curated a benchmark dataset for model training and release it to the public for others to study this new task and develop better models for table to image generation, as well as to leverage the trained models for other important and novel applications in practice.

- **Framework and Approaches:** We present a general framework for studying and developing approaches to solve the proposed data table to image generation task. Furthermore, we propose and evaluate a few different approaches for solving this task and extensively evaluate the quality and diversity of the generated images from such models across a range of different categories in our benchmark dataset.

- **Evaluation:** Experiments are designed to extensively evaluate the proposed models for a variety of data tasks. The proposed models along with the images generated are evaluated based on image quality and diversity of the images recommended.

## 2 PROBLEM FORMULATION

We aim to generate images describing the underlying segment/trend of the given table. To formulate this mathematically, let $\mathcal{L} \subseteq \Sigma^*$ be a language over some finite alphabet $\Sigma$ (e.g. think of $\Sigma$ consisting of all ASCII symbols) and let $y \in \mathcal{L}^{m \times n}$ be a tabular input with $m$ rows and $n$ columns over the language $\mathcal{L}$. For simplicity, we also mandate the first row of $y$, $y_{1:}$ to always be the column names.

Our goal can hence be described as, given a tabular input $y \in \mathcal{L}^{m \times n}$, we aim to generate an informative and high-quality image $\widehat{x} \in \mathbb{N}^{h \times w \times d}$, where $h, w, d$ is the height, width and depth that resembles real-world images and contains relevant information about $y$.

In order to achieve our goal, one possible approach is to assume that the real-world images $x$ and the tables $y$ follow a joint distribution $p(x, y)$, therefore given a table $y$, we can model the mapping rule $h : y \rightarrow x$ by the conditional distribution $p(x|y)$, which can be learned by constructing a training dataset $\{(y_i, x_i)\}_{i=1}^n$. After training on the dataset, we thus obtain a table-to-image model $\hat{h} : y \rightarrow x$ approximating the true conditional distribution, and we can then use it to infer a set of images for any new unseen tabular dataset $y_{\text{new}}$ as $\widehat{x} = \hat{h}(y_{\text{new}})$. We will implement and analyze this approach in detail in the later sections.

To evaluate the performance of approaches that generate images given tabular input, we adopt common metrics that measure the quality and diversity of the images generated, namely FID scores Heusel et al. (2017) and LPIPS Zhang et al. (2018). However, despite methods for quantifying the correlations between text and images has been proposed recently (e.g. CLIP (Radford et al., 2021)), it is not obvious how the relevancy between tables and images should be measured, since merely considering the textual component within tabular data risks losing structural information contained in the table, which is in fact a key component we aim to model during the image generation process in our work. Therefore, we simplify the connection between tables and images based on a simple rule to be described in the next section, and inspect relevancy qualitatively based on the visual perception of the image.

## 3 BENCHMARK DATASET

We benchmark the performance of table-to-image generation models based on a dataset consisting of 300 pairs of table and images. All images are downloaded from the Internet based on search results using keywords of common groceries, e.g. meat or fish; and each of the tables contains a subset of records found in Kaggle's open dataset "Marketing Campaign" (Saldanha, 2020), in which we only include the columns that indicates the amount of sweet, meat, fish, fruit, gold and wine purchased within a transaction by a consumer. The relevant columns are then standardized based on their column-wise mean and standard deviation, and we limit the resulting float number representation to 4 decimal points.

Next, we employ the idea of weak supervision (Ratner et al., 2016) in order to construct the dataset of table-image pairs. Weak supervision is a labeling technique that creates a data-label pair by

assigning labels based on a simplified mapping rule that can be done efficiently and automatically, instead of establishing the mapping manually, which can often be expensive and hard to obtain. Although the idea of weak supervision is first proposed to overcome the difficulty when dealing with insufficient training data for classification tasks, in this work we explore its possible application in image generation tasks and adopt its idea in constructing our benchmark dataset.

Hence, under this paradigm, we identify simple relations between a given table and an image based on the following rule: for a table $y \in \mathcal{L}^{m \times n}$ whose rows are sampled from the aforementioned Kaggle dataset and normalized column-wise, we assume the first row $y_{1:}$ consists of column names and $\langle y_{ij} \rangle \in \mathbb{R}$ for all $i \in \{2, ..., m\}$ and $j \in \{1, ..., n\}$, where the operator $\langle \cdot \rangle : \mathcal{L} \to \mathbb{R} \cup \{\bot\}$ extracts the numerical value of a string in the language $\mathcal{L}$ if the string represents a valid decimal number, otherwise it returns $\bot$. We then associate an image produced by a keyword $y_{1j}$ with the table $y$ if $\forall k \in \{2, ..., m\}, y_{kj} = \arg\max_{i \in \{1,2,...,n\}} \langle y_{ki} \rangle$ and define the column $j$ as the *pivot column* of table $y$; in other words, we say column $j$ is the pivot column if for every row $k$, $\langle y_{kj} \rangle$ is greater than that of any other columns in the same row, and an image produced using the pivot column name as the keyword is paired up with $y$.

With this procedure, we defined 300 table-image pairs across 6 categories: meat, wine, sweet, fish, gold, fruit, each with 50 table-image pairs. All images are resized to $256 \times 256$, and all tables consist of 5 to 20 rows.

## 4 APPROACH

In this section, we formally describe the proposed approaches for our new problem studied in this work. We discuss the advantages of each and begin by describing the simplest first. Within each subsections, we will also discuss simple variations of each methods and their possible improvements. Since generating images based on tabular data is a new problem that has not been investigated before, we start with straightforward composition of pre-existing models to gain better understanding of the key technical challenges for table-image generation, as well as setting a baseline for comparing the strengths and weakness of different approaches. We will also investigate the possibility of directly designing an end-to-end approach for table-image generation, and compare the results quantitatively and qualitatively against the simpler composite approaches.

### 4.1 DALLE-COLUMNNAME

For a given table $y \in \mathcal{L}^{m \times n}$ generated based on the process described before, we directly input the pivot column names to DallE (Ramesh et al., 2021; 2022) and select the top images generated based on their CLIP score with respect to the input column name. For example, given a table whose pivot column name is "MntFruits", we will directly input "MntFruits" into DallE to generate images.

### 4.2 DALLE-GPT3-CAPTION

Similar to the previous approach, we also pick the pivot column name from the input table. However, instead of directly feeding it into Dalle, we first input the pivot column name to GPT-3 (Brown et al., 2020) along with a textual description asking it to generate a suitable text prompt, and after retrieving its reply, we use it as the input to Dalle and finally select the top images with the highest CLIP score. For instance, with a table whose pivot column name is "MntWines", we input the following text to GPT-3: *Generate an image caption using "MntWines."* The GPT-3 model will then sample a reply from the underlying conditional distribution, say, *A bottle of wine on a mountaintop, with the caption "MntWines: The best wine for your next adventure."* We will then use its reply directly as the input to dalle-mini for image generation.

### 4.3 DALLE-TABBIE

Given the above approaches, it is natural to ask whether we can design an end-to-end process for the image generation process (Figure 2). Hence, based on DALL-E and DALLE-mini's model architecture (Ramesh et al., 2021), we design a tabular-data-to-image generation model based on two components: a VQGAN image encoder/decoder (Esser et al., 2021) and a BART sequence-to-sequence model (Lewis et al., 2019) that outputs image tokens auto-regressively given tabular

representations as inputs. We mathematically introduce the objective functions our model is trained to optimize in the following paragraphs.

Same as we have defined above, let $y \in \mathcal{L}^{m \times n}$ be a tabular input and $\widehat{x} \in \mathbb{N}^{h \times w \times d}$ be a generated RGB image with height $h$, width $w$ and color depth $d$. We also define $x \in \mathbb{N}^{h \times w \times d}$ as a real-world RGB image and $z \in \mathcal{Z}$ as the latent representation for an image. Hence, assuming the data generation process can be modeled by the Markov chain $x \rightarrow z \rightarrow y$, we can lower-bound the log-likelihood of $x, y, p(x, y)$ by

$$
\begin{aligned}
\log p(x, y) = \log \int_{z \in \mathcal{Z}} p(x, y, z) dz &= \log \int_{z \in \mathcal{Z}} p(x) p(z|x) p(y|z) dz = \log \mathbb{E}_{z|x} \Big[ p(x) p(y|z) \Big] \\
&\geq \mathbb{E}_{z|x} \Big[ \log p(x) + \log p(y|z) \Big] \\
&= \mathbb{E}_{z|x} \Big[ \log p(x) + \log p(y|z) - \log p(z|x) + \log p(z|x) + \log p(z) - \log p(z) \Big] \\
&= \mathbb{E}_{z|x} \Big[ \log p(x|z) \Big] - \mathrm{D}_{\mathrm{KL}}(p(z|x)||p(y, z)) \\
&\approx \mathbb{E}_{z \sim q_\psi(z|x)} \Big[ \log p_\theta(x|z) \Big] - \mathrm{D}_{\mathrm{KL}}(q_\psi(z|x)||p_\phi(z|y)p(y))
\end{aligned}
$$

where $\mathrm{D}_{\mathrm{KL}}$ is the KL-divergence between two probability distributions, $q_\psi, p_\theta$ is the encoder and decoder of VQGAN respectively, and $p_\phi$ is the BART decoder. The last line in the objective function above is commonly known as ELBO in variational inference literature, and in our model we approximate those distributions through common neural networks. However, in our problem setup we have multiple distributions to optimize, therefore we follow a similar training approach to that of DALL-E mini's by splitting the training process into 3 stages, which we describe in detail below.

**1. Fine-tuning VQGAN**    Same as previous works, we started with the VQGAN model trained on ImageNet (Deng et al., 2009). We fine-tuned our VQGAN model for 10 epochs to adapt it to our image dataset's distribution.

**2. Fine-tuning TABBIE**    Since our model crucially relies on tabular data as input, we need to produce a good representation for tables. In this work, we utilize TABBIE (Iida et al., 2021) in our model to extract relevant structural and textual/numerical information for the input table. TABBIE learns tabular representations through an objective known as corrupt cell detection, which asks the model to detect if any element within the table was replaced by an inconsistent one. The embeddings produced by TABBIE has shown to perform well on various tabular data mining tasks, therefore we utilized the pretrained representation as input to the BART encoder. To ensure the embedding vectors align with the tables we have, we fine-tuned TABBIE based on the default configurations.

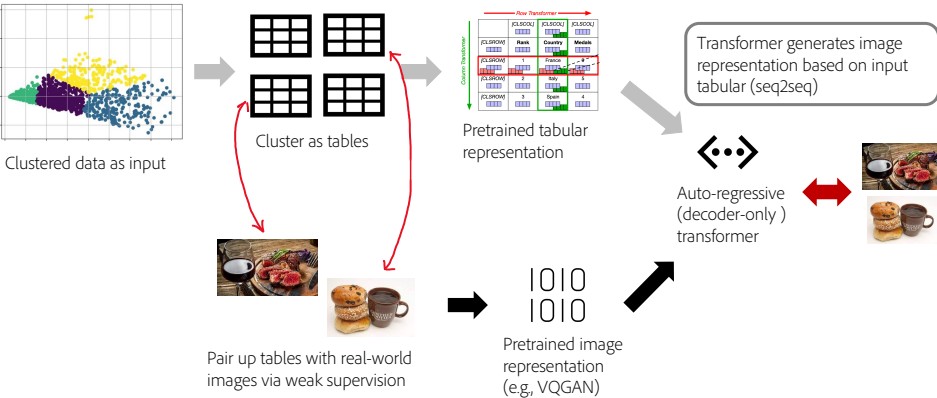

Figure 2: Overview of the end-to-end training process. See text for detailed discussion.

**3. Training BART**    Finally, based on the training procedure of DALL-E mini, we followed a similar approach for training the transformer. We preprocessed the training data by first encoding the images into tokens using VQGAN, and paired them up with their corresponding table's embedding vectors. The tabular embeddings are then passed into BART, aiming to produce an output sequence that matches the corresponding image. We trained the model for 30 epochs until the validation loss begins to deviate from the training loss. We also performed a grid search for the hyperparameters used in sampling from the conditional distribution for image generation by maximizing the CLIP score of the generated images with respect to their pivot column names.

## 4.4    Discussion

It is also possible to generate images following a similar methodology when textual input is given, this includes directly inputting the table into Dalle-mini, asking GPT-3 to generate a caption given the tabular input, and a more comprehensive approach that trains a tabular-image joint representation similar to CLIP. However, how to define a good paring rule between a table and an image remains the main challenge for producing a model that generates meaningful images.

Aside from the difficulty of data preparation and bechmarking, since we have only focused on dealing with tables that mostly contain numerical information, it is also worth investigating for the end-to-end model approach, whether a latent representation, especially those trained based on tokenization the input first actually encodes those numerical trends. In fact, it will also be interesting if we can directly use the numerical values within the table as its embedding vectors, given that we have identified a reasonable method to ensure the consistency of the embedding dimensions.

However, as we mentioned above, we currently limit the input table to only contain numerical data, and it is still unclear what we should expect if other data type are also present in the input table. Furthermore, even for our current task, it is also worth asking if we can do anything beyond merely generating an image of an item, such as including persons or even backgrounds that also reflects some information about the input table. We leave these possible improvements as part of our future directions.

## 5    Related Work

### 5.1    Text-based Image Generation

Recently, there has been a lot of work towards generating images from textual prompts. Some of the most famous results are those of DALLE Ramesh et al. (2021) and DALLE-2 Ramesh et al. (2022). The model architecture of DALLE consists of an image encoder/decoder and a transformer, and it learns to generate images by concatenating the textual input token along with the image tokens by modeling it as a sequence-to-sequence task, thus taking advantage of the powerful capabilities of transformers to produce an image as the output sequence. On the other hand, DALLE-2 leverages CLIP embeddings that jointly model text/image pairs for image generation. There are also other works (Saharia et al., 2022; Rombach et al., 2022; Nichol et al., 2021) that investigates text-to-image generation methods. However, all of these works have focused on generating images from text-based prompts whereas we focus on image generation from numeric tabular data.

### 5.2    Tabular Data Representation Learning

There has been a lot of recent work focused on developing models for learning representations from tabular data to perform various downstream tasks, including trend detection and tabular question-answering. We highlight a recent approach known as TABBIE (Iida et al., 2021) that learns the embedding of tables by training on the objective of detecting corrupted cells. One notable results of the embdiings learned by TABBIE is that for identifying numeric trends in the input table, it outperforms multiple previous works and demonstrates the potential of applying these embeddings to more complex tasks. Nonetheless, all of these approaches have not focused on our specific task of generating an image or set of images from a given tabular dataset.

To the best of our knowledge, this paper is the first to introduce this task, curate a benchmark dataset for it, develop techniques for solving it, and evaluating the approaches both in terms of quality and diversity of the images generated by them from the tabular datasets.

Table 1: Image quality (FID) and diversity (LPIPS) results.

| Data | Model | Image Quality (FID) ↓ | Image Diversity (LPIPS) ↑ |
|---|---|---|---|
| Gold | DALLE-ColumnName | $415.29 \pm 18.18$ | $0.75 \pm 0.05$ |
| | DALLE-GPT3-Caption | $\mathbf{335.93} \pm 16.53$ | $0.72 \pm 0.01$ |
| | DALLE-Tabbie | $701.04 \pm 29.43$ | $\mathbf{0.77} \pm 0.06$ |
| Wine | DALLE-ColumnName | $295.94 \pm 15.22$ | $0.77 \pm 0.04$ |
| | DALLE-GPT3-Caption | $\mathbf{277.66} \pm 19.42$ | $0.79 \pm 0.03$ |
| | DALLE-Tabbie | $338.90 \pm 33.54$ | $\mathbf{0.86} \pm 0.07$ |
| Meat | DALLE-ColumnName | $492.55 \pm 33.32$ | $0.78 \pm 0.03$ |
| | DALLE-GPT3-Caption | $\mathbf{358.04} \pm 29.98$ | $0.72 \pm 0.05$ |
| | DALLE-Tabbie | $515.35 \pm 46.32$ | $\mathbf{0.81} \pm 0.06$ |
| Fruit | DALLE-ColumnName | $\mathbf{354.21} \pm 20.61$ | $\mathbf{0.82} \pm 0.03$ |
| | DALLE-GPT3-Caption | $426.27 \pm 22.28$ | $0.79 \pm 0.02$ |
| | DALLE-Tabbie | $359.18 \pm 38.66$ | $0.78 \pm 0.05$ |
| Sweet | DALLE-ColumnName | $\mathbf{327.84} \pm 17.10$ | $0.78 \pm 0.04$ |
| | DALLE-GPT3-Caption | $465.79 \pm 23.07$ | $0.70 \pm 0.04$ |
| | DALLE-Tabbie | $437.07 \pm 43.15$ | $\mathbf{0.83} \pm 0.07$ |
| Fish | DALLE-ColumnName | $\mathbf{506.60} \pm 12.57$ | $0.73 \pm 0.02$ |
| | DALLE-GPT3-Caption | $509.78 \pm 13.66$ | $0.74 \pm 0.03$ |
| | DALLE-Tabbie | $530.37 \pm 26.70$ | $\mathbf{0.78} \pm 0.06$ |

## 6 EXPERIMENTS

To understand the utility and effectiveness of the proposed approaches for this new problem, we designed experiments to quantitatively evaluate the quality and diversity of the images generated from the various approaches. Furthermore, we also show the actual images generated from the proposed models and provide a detailed discussion on some of the interesting insights observed.

### 6.1 EXPERIMENTAL SETUP

For each approach, we compute its FID with respect to the benchmark dataset as follows: for DALLE-ColumnName and DALLE-GPT3-Caption, we randomly partition the 20 images into 4 groups, each with 5 images, and we measure the FID between the 5 images within each partition against all images in the benchmark dataset under the same category; and for DALLE-Tabbie, we also randomly partition the 200 images into 4 groups and measure their FID with respect to the benchmark dataset. Note that since we observed the variance of DALLE-Tabbie to be typically larger than the other approaches, we reported results of 200 generated images. Nevertheless, similar performance was observed when using only 20 images as well. To measure LPIPS (Zhang et al., 2018), we simply pair up all the images produced by each approach with those in the benchmark dataset and compute the mean and variance of the resulting scores.

### 6.2 QUANTITATIVE RESULTS

We quantitatively compare the quality and diversity of the images generated by the proposed models. For this, we use the FID (Heusel et al., 2017) evaluation metric to quantify the quality of the images and leverage LPIPS (Zhang et al., 2018) to evaluate the diversity of the images generated from the proposed models. Results for various data in our benchmark corpus are reported in Table 1. We observe that the model generating the highest quality images highly depends on the underlying data category from the benchmark corpus. Notably, DALLE-GPT3-Caption performed best for Wine,

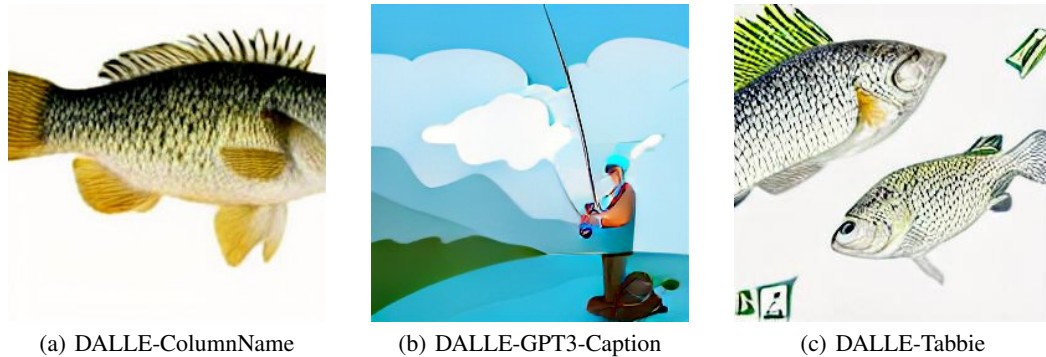

| (a) DALLE-ColumnName | (b) DALLE-GPT3-Caption | (c) DALLE-Tabbie |

Figure 3: Qualitative comparison of the images generated from the different approaches.

Meat, and Gold in terms of the quality of images generated whereas DALLE-ColumnName generated the highest quality images for Fruit, Sweets, and Fish. Intuitively, Wine and Meat are often purchased together, and thus this makes sense. Conversely, fruits and sweets are also related. Furthermore, in nearly all cases, DALLE-Tabbie generates the images with largest diversity. The only exception is for the fruit dataset where DALLE-ColumnName achieves the best diversity. In this case, DALLE-Tabbie generates images of very high quality, outperforming DALLE-GPT3-Caption, and relatively close to the quality of DALLE-ColumnName. Hence, we observe a trade-off between image quality and diversity.

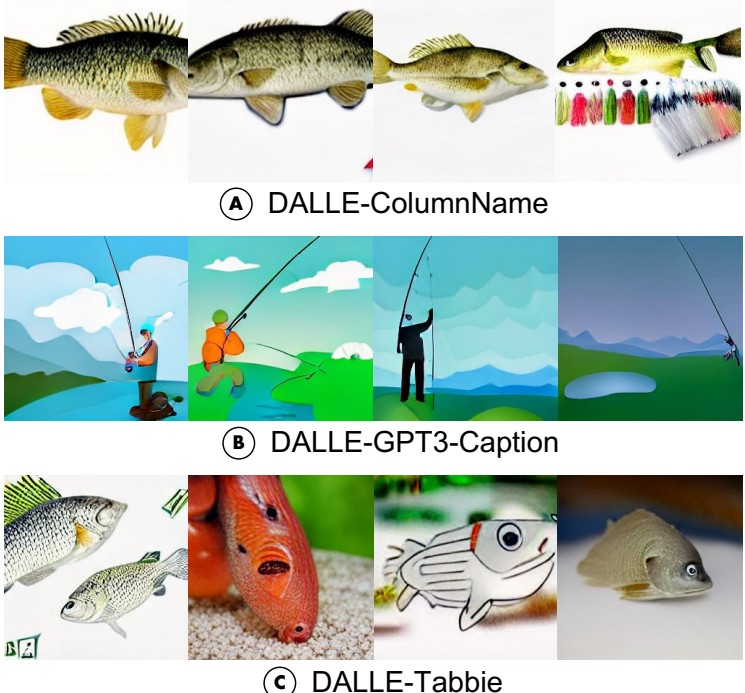

Figure 4: Comparison of the diversity of the images generated by the proposed approaches.

## 6.3 QUALITATIVE RESULTS

We now compare the images generated from the approaches qualitatively in Figure 3. In particular, we show images from the fish data generated from each of the approaches described in Section 4. Interestingly, we see that DALLE-GPT3 generates a painting-like image of a person fishing on a cloudy day with mountains behind them. However, DALLE-ColumnName generates a more realistic

image of a fish whereas DALLE-Tabbie generates an image seemingly in-between these two extremes in the sense that the fish is realistic looking with some artistic features such as the dorsal fin of the fish near the top of the generated image in Figure 3. Furthermore, the image genertaed by DALLE-Tabbie also contains two fish facing in opposite directions. In Figure 4, we investigate the diversity of the images generated by the proposed approaches. Notably, the images generated by DALLE-ColumnName are all very similar showing a fish with mostly the same features. Similarly, DALLE-GPT3-Caption generates images with a very similar scene that includes an individual fishing with mountains in the background. All of the images have very similar colors as well. Both DALLE-ColumnName and DALLE-GPT3-Caption suffer in terms of the diversity of images that these models are capable of generating. However, DALLE-Tabbie is able to generate images that are significantly more diverse as shown in Figure 4. Notably, the images are even diverse in terms of style, that is, realistic images as well as those that appear to be drawn, to even those that are a hybrid with some realistic and artistic features and colors.

## 7 CONCLUSION

In this work, we considered the problem of tabular-data-to-image generation motivated by various real world applications and investigated multiple potential approaches to achieve this task, and to the best of our knowledge this work is the first to study such problem. We created a benchmark dataset in order to evaluate the performance of different methods and construct an end-to-end model for image generation. We identified a framework for developing approaches to solve the proposed problem, and we evaluated the performance of those approaches quantitatively and qualitatively.

We conclude that future work should develop more sophisticated approaches for this new task and explore using such techniques for other exciting applications. Furthermore, by releasing an initial benchmark dataset for this important new learning task in order to facilitate others to work on this problem and develop new approaches to solve it, we hope that future work will expand and enhance this initial benchmark dataset with other types of tabular datasets and corresponding image pairs.

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
