# OpenReview forum: "Tabular Data to Image Generation: Benchmark Data, Approaches, and Evaluation"
_ICLR.cc/2023/Conference — Submitted to ICLR 2023_

### Official Review · Reviewer_PnA1 · 2022-10-25

**Confidence:** 4
**Correctness:** 3
**Technical Novelty And Significance:** 3
**Empirical Novelty And Significance:** 3
**Recommendation:** 3

**Clarity, Quality, Novelty And Reproducibility:**

Please see the weakness part.


**Strength And Weaknesses:**

The main idea of the paper is interesting. But the motivation what we could benefit from the generated images is not very clear.

There are some questions:
1. The authors need to provide more examples when the generated images help. The reviewer is confusing that we can generate an image for each sample or only for a whole set of tabular data.
2. The authors may consider providing more illustration examples in the paper. The current illustration images seem to be subjective.
3. Does the ability of the method depend on the quality of the pre-trained model?
4. In some real-world tasks, we may not know the exact meaning of the column, and it is difficult to find the pre-trained model in the specific area. For example, if the table is related to housing prices or climate change, could the authors show the corresponding images?

**Summary Of The Paper:**

This paper works on an interesting topic of generating images based on tabular data. It provides a benchmark dataset and proposes some approaches with varying levels of complexity and tradeoffs.


**Summary Of The Review:**

The paper is interesting, but there are still some concerns. The authors may try to address the issues and make the paper clearer.

---

### Official Review · Reviewer_CFxB · 2022-10-27

**Confidence:** 4
**Correctness:** 2
**Technical Novelty And Significance:** 2
**Empirical Novelty And Significance:** 2
**Recommendation:** 3

**Clarity, Quality, Novelty And Reproducibility:**

Clarity
- Figures and explanations are generally difficult to follow
- Problem statement is very strong and well defined

Quality
- The figure's quality is very low and - in at least one case a component is directly copy-pasted from another paper.

Novelty
- The proposed problem is original, and the authors create original baseline models that combine existing modeling approaches

Reproducibility
- The models are described and a reader could likely reimplement the pipelines, the data is also mentioned to be released but I could not find it


**Strength And Weaknesses:**

Strengths
- [major] Problem formulation is very understandable
- [major] New modeling approaches for end-to-end conditional generation

Weaknesses
- [major] Not nearly enough results and qualitative output, the paper is about the value of visually summarizing tabular data but there is minimal to no qualitative output
- [major] Figures are out of place and are relatively difficult to follow, (ex: components are unrelated in figure 1 and are actively confusing)
- [major] Figure from TABBIE paper seems to be directly copied and pasted onto one of your figures
- [minor] Benchmark dataset is described, but very little information is given (ex: what size are the tables, distribution of categories)
- [minor] usefulness/applications are very simply motivated and then rarely addressed when they are a big component of the problem.

**Summary Of The Paper:**

The paper introduces the problem of mapping from table data to images. The author's test strategies directly generate images from tabular data (which can include numerical values). This would give an end user the ability to create a visual summary of the data they are looking at. Generated images are evaluated using FID and LPIPS, and results for methods are additionally qualitatively discussed.

**Summary Of The Review:**

I think the idea is interesting from an application perspective, and the problem is very well defined by the authors- specifically the problem of generating a "visual summary" of table data. But throughout the paper, there is not enough detail on the data that the models are being trained on. Additionally,  there is not nearly enough qualitative output - which would be expected for a paper that aims to do conditional generation. When figures exist are generally hard to follow and sometimes actively confusing (ex: components of Figure 1).

No evaluation in the paper seems to critically address one big component of the paper - do the resulting images well summarise the table data in some meaningful way? While I understand this is very difficult to quantify, there is zero to minimal discussion on this point. Since this is a conditional generation problem, do my generated images in some way tell me about the data they are generated from? A good solution to this problem would not just generate images that are not only high quality, which your metrics measure, but also relevant to the data they are conditioned on, and I'm not sure that this is addressed in any detail.

---

### Official Review · Reviewer_Leo6 · 2022-10-31

**Confidence:** 4
**Correctness:** 2
**Technical Novelty And Significance:** 1
**Empirical Novelty And Significance:** Not applicable
**Recommendation:** 3

**Clarity, Quality, Novelty And Reproducibility:**

- The quality of writing is poor.
- Although the tabular-data-to-image generation is new and unexplored, this paper lacks novel designs to address it.
- It's hard to reproduce the proposed approach based on the current version because Section 6.1 Experimental Setup only mentioned little key hyperparameters. Also, this paper mentions "release the proposed benchmark" many times, but this work neither uploads the dataset in the supplementary materials nor shares an anonymous GitHub repository.

**Strength And Weaknesses:**

### Strengths

- This work studies a novel and somewhat interesting problem, i.e., tabular-data-to-image generation.
- This paper curates a new benchmark dataset for the tabular-data-to-image generation problem.



### Weaknesses

- The quality of writing is poor. It is hard to follow this paper.
- The technical contributions are limited. This work simply applies prior techniques, e.g., DALL-E, GPT3, VQGAN, and TABBIE, to the tabular-data-to-image generation, lacking novel technical designs.
- The experimental evaluation is weak. The results in Section 6 are insufficient to support the main claim in the Abstract, that is, "The set of generated images provides an intuitive visual summary of the tabular data".
- In Section 6.1 Qualitative Results, this paper mainly discusses the quality of generated images. However, from my point of view, the correlation between the tabular input and the generated images is more worthwhile to discuss.
- The proposed benchmark only contains 300 pairs of tables and images with a limited number of categories. I don't think this is sufficient to train a generative model for real-world tabular-data-to-image generation problems.

**Summary Of The Paper:**

This paper addresses the tabular-data-to-image generation problem, which is novel and somewhat interesting. To study this unexplored problem, this work first curates a benchmark dataset containing 300 pairs of tables and images. Then, this paper utilizes existing models, e.g., DALL-E, GPT3, VQGAN, and TABBIE, to generate images from the tabular data. Finally, the authors qualitatively and quantitatively evaluate the proposed method.

**Summary Of The Review:**

Although this work addresses a novel and interesting tabular-data-to-image generation problem, the quality of writing is poor, the technical contributions are limited, and the experimental evaluations are weak. Based on my aforementioned concerns, I recommend rejecting this paper.

---

### Decision · Program_Chairs · 2023-01-20

**Decision:**

Reject

**Justification For Why Not Higher Score:**

All reviewers felt the paper should be rejected and the authors did not submit a rebuttal.

**Justification For Why Not Lower Score:**

N/A

**Metareview: Summary, Strengths And Weaknesses:**

The paper studies the problem of generating images from tabular data, and introduces a new dataset of (table, image) pairs to facilitate this work. Reviewers felt that the problem itself was novel and potentially interesting, but that the overall quality of the writing in the paper was poor. Reviewers also felt that the paper had limited technical novelty compared to prior work on generative models, and that the experimental evaluation was not convincing. The paper also did not provide enough detail about the dataset on which models were trained. The authors did not submit a rebuttal, and in the end all reviewers felt that the paper was not ready for publication.